# Where octagonal geometry meets chaos: A new S-Box for advanced cryptographic systems

Abdulbasid Banga[1], Yasir Mahmood[2], Naif Al Mudawi[3], Nisreen Innab[4], Nadeem Iqbal[5]*, Hossam Diab[6,7]

1 Saudi Electronic University, College of Computing and Informatics (CCI), Riyadh, Saudi Arabia, 2 Department of Computer Science & Software Engineering, College of IT, United Arab Emirates University (UAEU), AlAin, Abu Dhabi, United Arab Emirates, 3 Department of Computer Science and Information System, Najran University, Saudi Arabia, 4 Departmentof Computer and Information Systems, College of Applied Sciences, AlMaarefa University, Diriyah, Riyadh, Saudi Arabia, 5 Department of Computer Science & IT, The University of Lahore, Lahore, Pakistan, 6 Menoufia University, Math and Computer Science Department, Faculty of Science, Menoufia, Egypt, 7 Taibah University, Computer Science Department, Applied College, Madinah, Saudi Arabia

☯ These authors contributed equally to this work.
* nadeem.iqbal537@gmail.com

**Data availability statement:** All relevant data are within the manuscript.

## Abstract

Substitution Box (S-Box) has had been a cardinal component of various cryptographic systems. In this paper, we introduce a novel S-Box design that merges octagonal geometry with chaotic dynamics to enhance the security effects of the cryptographic systems. In particular, the proposed method leverages the geometric properties of octagons and the unpredictability of chaotic maps to construct a novel S-Box with improved security features. The mathematical construct octagon carries out the necessary operation of confusion in the proposed S-Box. The centres of these octagons are hypothetically created within the confines of the $16 \times 16$ matrix of numbers. Further, these octagons have different radii, locations, and the amounts with which the numbers lying on their boundaries have to be circularly shifted clockwise or anti-clockwise to create the confusion effects. In case, a portion of octagon goes past the edges of the matrix, the numbers lying on its boundary have been wrapped out. This process has been repeated numerous times to come up with a reliable and a secured S-Box. The comprehensive security analyses validate that the proposed S-Box is furnished with nice security effects and has the requisite resilience to defy the varied cryptanalytic threats. The results of non-linearity and differential probability are 105.625 and 0.0391 respectively which signals towards the inherent robustness of the suggested S-Box.

## 1 Introduction

In the rapidly evolving landscape of cybersecurity, the need for robust cryptographic mechanisms is more pressing than ever in the history of mankind. At the heart of many cryptographic algorithms lies the Substitution box, or S-Box [4], the horsepower of cryptography

**Competing interests:** The authors have declared that no competing interests exist.

that cements the security and integrity of varied encryption processes. Although it is a small element but it plays a very cardinal role in safeguarding sensitive information across numerous domains, including communication [5], finance [28], healthcare [6], and smart cities [7] to name a few.

The block ciphers like Advanced Encryption Standard (AES) [8] and Data Encryption Standard (DES) [9] depend heavily over the S-Box. These ciphers carry out substitution operations to introduce non-linearity into the encryption process. Non-linearity is very important since it protects encrypted data from differential and linear cryptanalysis, making it exceedingly difficult for attackers to predict or reverse-engineer the encrypted information.

In the various communication systems, the role of S-Boxes is critical. It safeguards the integrity and confidentiality of data [1]. By permuting the plaintext in a way which is computationally infeasible to reverse without having the knowledge of the correct key, S-Boxes facilitate in protecting the sensitive communications, whether they are between individuals, organizations, or even between devices in the Internet of Things (IoT) [10]. Open and insecure communication, on the other hand, may result in eavesdropping, data breaches, and unauthorized access, but robust S-Box facilities mitigate these risks efficiently.

The financial sector depends upon the enterprise of cryptography for the secure transactions, authenticating users, and protecting other digital assets [11]. S-Boxes play a key role in encrypting the financial data, like credit card information, online banking transactions, and blockchain-based cryptocurrencies [12]. The robustness of an S-Box can directly impact the safety of financial systems, making it a critical component in preventing fraud, identity theft, and financial cyberattacks.

With the increasing digitization of medical records and the rise of telemedicine, the healthcare sector is particularly vulnerable to cyber threats [13]. S-Boxes are employed in the encryption algorithms that protect electronic health records (EHRs), ensuring that patient data remains confidential and inaccessible to unauthorized parties [14]. The robustness of the S-Boxes used, can be the difference between secure patient data and a devastating breach that compromises sensitive health information.

As smart cities evolve, they integrate vast networks of interconnected devices and systems, from traffic management and energy grids to surveillance and emergency response systems. These systems are prime targets for cyberattacks [15]. S-Boxes contribute to the security of the cryptographic protocols that protect the data flowing through these systems, ensuring that critical infrastructure remains resilient against cyber threats. In a smart city, where the stakes are high, the reliability and strength of the S-Boxes used, can have far-reaching implications for public safety and national security [16].

In the recent years, many efforts have been made to develop S-Boxes based on numerous constructs and theories like algebra [17–21], chaotic systems [22–24], trigonometry [25], complete latin square [4], cuckoo search algorithm [26], chess piece Castle [1] etc. The work [17], for example, exploited the inherent power of adjacency matrices and graph theory in order to design a novel S-Box algorithm. In particular, the authors of the reported work employed the coset diagram for action of the Mobius group. Besides, the theory of Galois field $GF(2^8)$ was also employed to render novel S-Boxes. Moreover, the work [19] used the elements of the multiplicative subgroup of the Galois field instead of the entire Galois field to write novel S-Box. The thorough security analysis of the said work rendered very promising results. Additionally, the newly developed S-Box was applied for watermarking of digital images which produced nice results.

In a yet another research endeavor [22], novel S-Box was written using the chaotic Rabinovich–Fabrikant fractional order (FO) system. Additionally, a new key-based permutation technique was developed to enhance the functionality of the initial S-Box to

construct the final S-Box. The security analysis demonstrated that the proposed S-Box was more secured and robust as compared to the state of the art. Apart from that, final S-Box's efficacy in the image encryption was measured through the benchmark of majority logic criterion (MLC) which rendered very exceptional results. Moreover, the work [4] mixed the constructs of chaotic map and complete Latin square to come up with a novel S-Box algorithm. The peculiar modus operandi of the reported research work goes like this. Firstly, a complete Latin square was produced through the usage of chaotic sequences. Secondly, S-Box was constructed by using the complete Latin square. Validation of the reported work demonstrated that the suggested S-Box was furnished with high performance and has the power to defy varied cybersecurity threats. Moreover, the S-Box was applied over the digital images to show its applicability. Security analysis gave very satisfactory outcomes.

The current study introduces a novel algorithm for the construction of an S-Box. In particular, it fuses octagonal geometry with chaotic dynamics. To put this in other words, this study introduces a new class of S-Boxes that harnesses the peculiar geometric properties of octagons and the intrinsic randomness of chaotic maps. By the implementation of octagonal structures with a set of varied parameters, diffusion properties have been enhanced. Moreover, transformation through the chaos introduces additional layers of chaoticity and irregularity.

The motivation to address the issues with conventional S-Box designs, which could be vulnerable to the varied cryptanalytic threats, is what spurred this research. While chaotic maps render abundant non-linearity and randomness, octagonal geometry provides a distinctive setting for adjusting the substitution values. By synergizing these components, the goal is to produce an S-Box that outshines the services given by conventional cryptography methods.

In the sections that follow, we detail the preliminary studies (Section 2). In this section, Lorenz chaotic system and geometrical figure octagon have been described. In Section 3, the proposed algorithm using the construct octagon has been described in details. Simulation and performance analysis have been carried out in the Section 4. The results obtained and the particular methodology of the algorithm have been discussed in the Section 5. Sections 6 and 7 close the paper with necessary concluding remarks and the possible future work.

## 2 Preliminaries

### 2.1 Lorenz chaotic system

Edward Lorenz in 1960 came up with a novel chaotic system dubbed as Lorenz system after his name. Nonlinear system of ordinary differential equations written below characterizes this chaotic system. Apart from that, it is dynamical system [2]:

$$
\begin{aligned}
\dot{x} &= \sigma(y - x), \\
\dot{y} &= x(\rho - z) - y, \\
\dot{z} &= xy - \beta z
\end{aligned}
\tag{1}
$$

In the above set of equations, the variables $\sigma$, $\rho$ and $\beta$ correspond to the control parameters. Besides, state variables of this dynamical system are $x,y,z$. Using Runge–Kutta methods such as the RK45 [3], System (1) is commonly solved numerically. Additionally, other methods can also be applied. The chaotic attractors of both the 2D and 3D have been drawn in Figs 1–4.

### 2.2 Octagon —A mathematical construct

**Definition:** In geometry, an *octagon* is a polygon with eight sides, also known as an 8-gon. A *regular octagon* is a specific type of octagon where all sides are equal in length and all internal

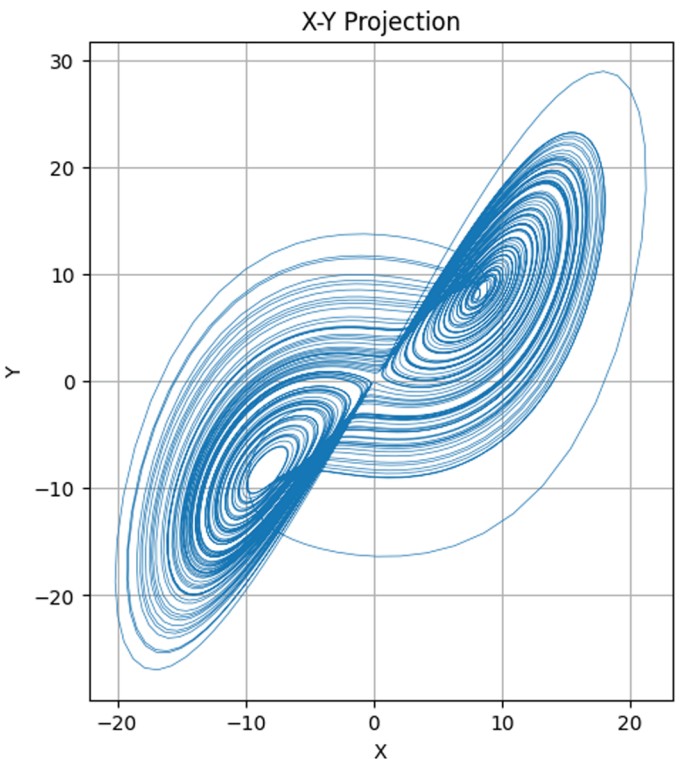

**Fig 1. Attractors in XY plane.**

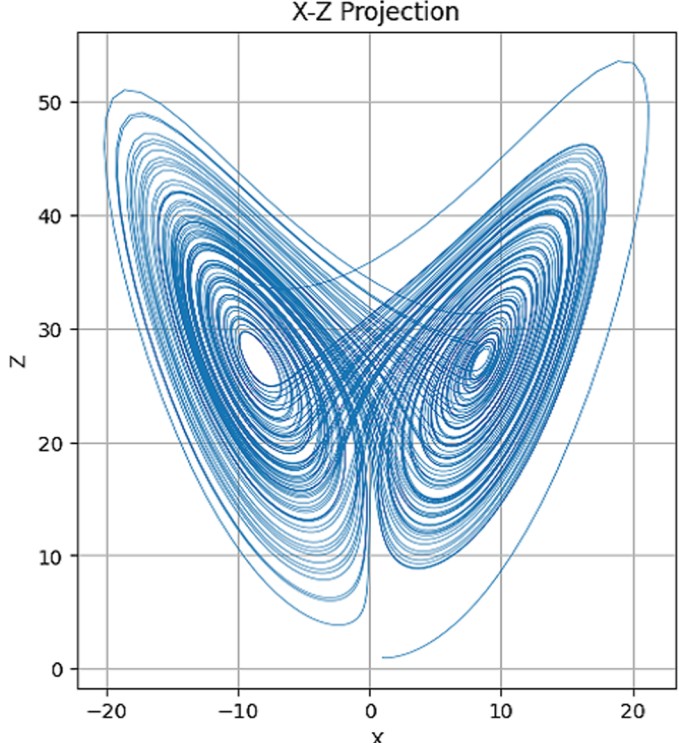

**Fig 2. Attractors in XZ plane.**

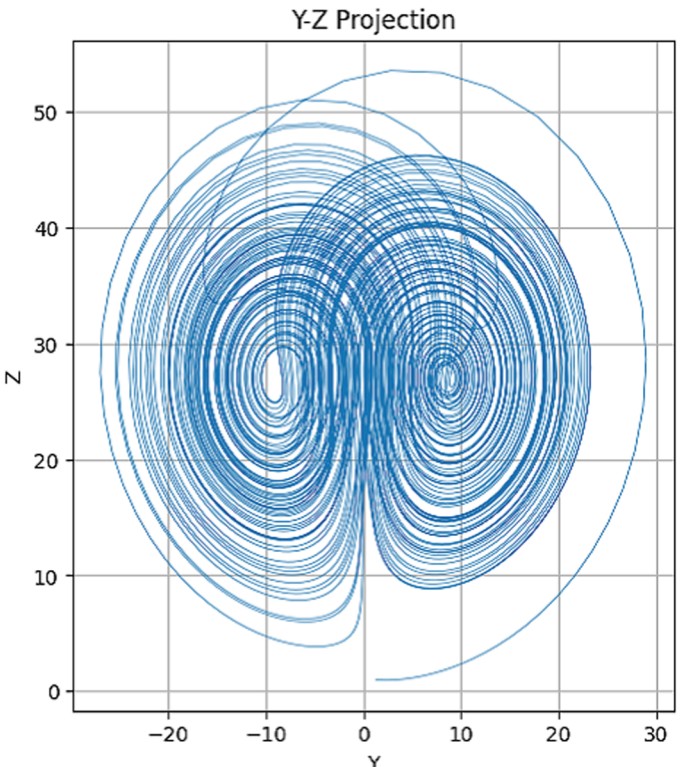

**Fig 3. Attractors in YZ plane.**

angles are congruent (see Figs 5–7). It possesses eight lines of reflective symmetry and exhibits rotational symmetry of order 8. Each internal angle in a regular octagon measures $135°$ (or $\frac{3\pi}{4}$ radians), while the central angle is $45°$ (or $\frac{\pi}{4}$ radians).

The vertex coordinates for a regular octagon, which is centered at the origin and has a side length of $s$, are:

- $\pm 1, \pm(1 + \sqrt{s})$
- $\pm(1 + \sqrt{s}), \pm 1$

In a regular octagon with side length $s$, there are three distinct types of diagonals, each with its own length, as described below:

- **Short diagonal:** $s\sqrt{2 + \sqrt{2}}$
- **Medium diagonal:** $(1 + \sqrt{2})s$ (silver ratio times $s$)
- **Long diagonal:** $s\sqrt{4 + 2\sqrt{2}}$

The inradius $r$, circumradius $R$, and area $A$ of a regular octagon can be directly calculated using the formulas for a general regular polygon with side length $s$, as follows:

$$r = \frac{1}{2}s\cot\left(\frac{\pi}{8}\right) \tag{2}$$

$$= \frac{1}{2}(1 + \sqrt{2})s \tag{3}$$

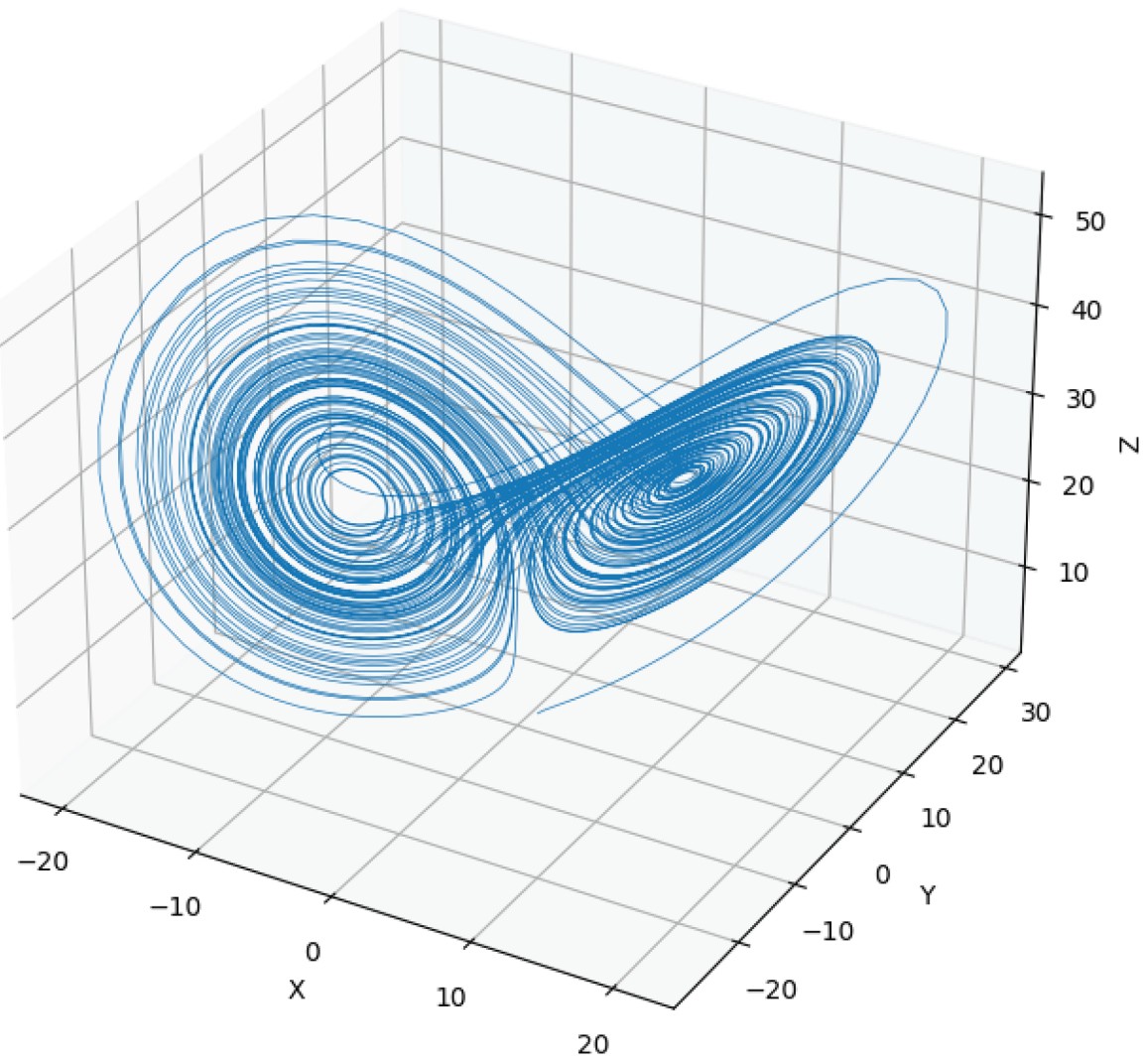

**Fig 4. Attractors in XYZ space (3D).**

$$R = \frac{1}{2} s \csc\left(\frac{\pi}{8}\right) \tag{4}$$

$$= \frac{1}{2} \sqrt{4 + 2\sqrt{2}} s \tag{5}$$

$$A = 2s^2 \cot\left(\frac{\pi}{8}\right) \tag{6}$$

$$= 2(1 + \sqrt{2}) s^2 \tag{7}$$

The central angle $\theta$, vertex angle $\alpha$, and exterior angle $\gamma$ are given by:

$$\theta = \frac{1}{4}\pi \tag{8}$$

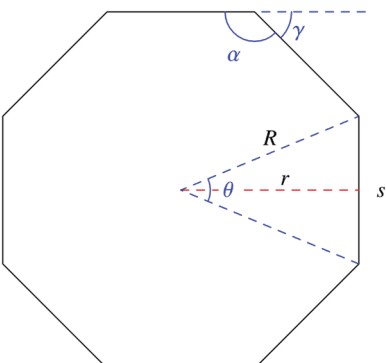

**Fig 5. An octagon with central angle $\theta$, interior angle $\alpha$, exterior angle $\gamma$, inradius $r$, circumradius $R$ and side $s$.**

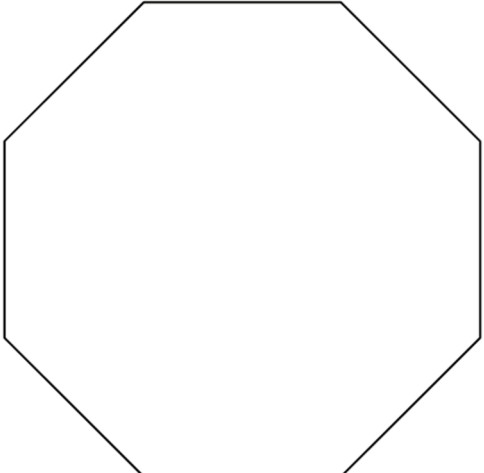

**Fig 6. A white-colored octagon with alternate sides parallel to the x- or y-axes.**

$$\alpha = \frac{3}{4}\pi \tag{9}$$

$$\gamma = \frac{5}{4}\pi \tag{10}$$

An octagon with an inradius of 1 can be specified by the following inequalities:

$$-1 < \frac{x-y}{\sqrt{2}} < 1 \wedge -1 < \frac{x+y}{\sqrt{2}} < 1 \wedge -1 < x < 1 \wedge -1 < y < 1 \tag{11}$$

An octagon may also be oriented so that its vertices lie along the $x$- and $y$-axes and at 45-degree angles to them, as illustrated in Fig 8. An octagon with circumradius $\frac{\sqrt{2}-1}{2}$ in this orientation can be specified by the inequality:

$$2(|x| + |y|) + \sqrt{2}\,(|x-y| + |x+y|) < 1 \tag{12}$$

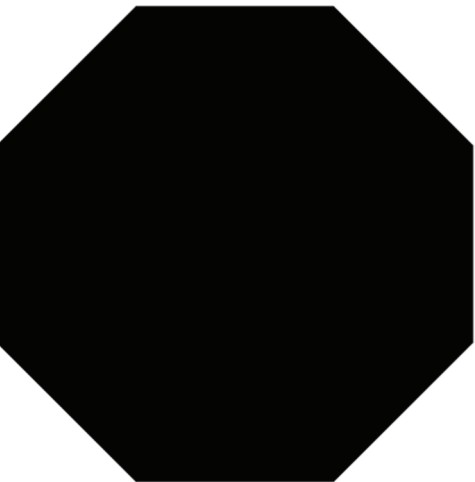

**Fig 7. A black-colored octagon with alternate sides parallel to the x- or y-axes.**

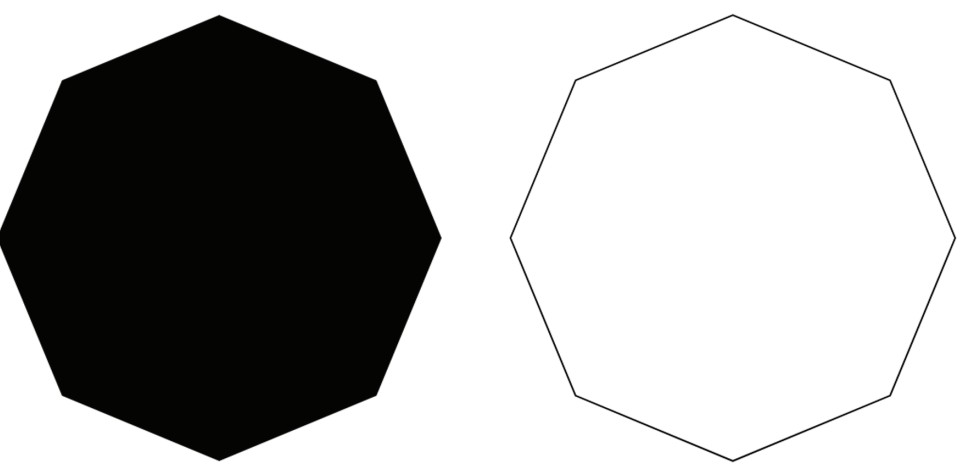

**Fig 8. Two octagons oriented at 45-degree angles along the *x*- and *y*-axes.**

**Relevance to Cryptography**. We contend that the unique geometric properties of an octagon can be leveraged to design a secure and efficient S-Box. Sliding boundary values around an octagon can introduce complex permutations, enhancing the cryptographic strength of the S-Box.

**Application in the Proposed Algorithm**. The construct of octagon will be used in the proposed algorithm, such as by creating boundaries of the octagon within the S-Box to scramble and reorder elements. This act will cement the security, including increased resistance to numerous cryptanalytic attacks.

## 3 Proposed octagonal chaotic transformation algorithm for S-Box (OCTA-SBox)

The mathematical figure octagon in the context of the new S-Box algorithm, has been employed in providing a dynamic scrambling mechanism. The S-Box experiences non-linear

permutations due to the systematic shifting of the octagon's border values. This technique makes use of the octagon's geometric features to improve confusion, which is one of the most important parts of secure encryption. The method obtains a greater level of security by using this shape, increasing its resistance to several types of cryptanalysis, including differential and linear assaults. The use of octagonal structures in the S-Box design offers several advantages:

- *Complex Permutations*: The shifting of octagonal boundaries allows for intricate permutations of data, significantly complicating potential attacks.
- *Increased Resistance*: The non-linear nature of the transformations introduced by the octagon enhances the algorithm's resistance to varied cryptanalytic techniques.
- *Efficiency*: The octagon's symmetry ensures that these transformations can be performed efficiently, maintaining the algorithm's performance while boosting security.

### 3.1 Key streams generation algorithm

This algorithm generates the key components for a cryptographic system by utilizing the Chaotic System (1). The input parameters of this algorithm are $x_0, y_0, z_0, \sigma, \rho, \beta, \psi$. The list of parameters $x_0, y_0, z_0, \sigma, \rho, \beta$ will spark the chaotic system while $\sqrt{\psi} \times \sqrt{\psi}$ refers to the size of the required S-Box. Output of the algorithm is the two key streams *octagon_centers*, *radius_steps*. The first key stream *octagon_centers* contains the coordinates of the octagon centers, while the second key stream *radius_steps* has the radii and the step values for each generated octagon. Fig 9 shows the different steps of this algorithm.

Here a step by step explanation will be given for this algorithm.

**Initialization:**

Line 1 sparks the chaotic system with the given values and parameters. It generates sequences $x$, $y$, and $z$ with length $\psi^2 + n_0$, where $n_0 \geq 500$. We have neglected the initial $n_0$ values of the keystream. The purpose of neglecting these values is the avoidance of the transient effects of the system.

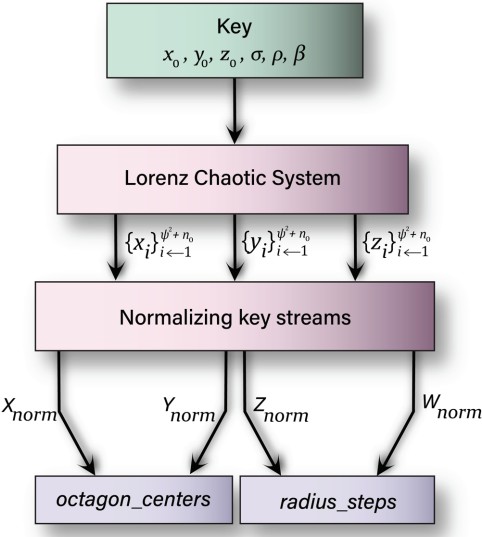

**Fig 9. Generation of key streams.**

**Algorithm 1.** *KeyStreamGeneration.*

```
Input: x₀, y₀, z₀, σ, ρ, β, ψ
```
$$\textbf{Input: } x_0,\ y_0,\ z_0,\ \sigma,\ \rho,\ \beta,\ \psi$$
$$\textbf{Output: } octagon\_centers,\ radius\_steps$$

1: Spark the Chaotic System (1) by providing it the initial values $x_0$, $y_0$, $z_0$ and the system parameters $\sigma$, $\rho$, $\beta$ to generate $\{x_i\}_{i \leftarrow 1}^{\psi^2 + n_0}$, $\{y_i\}_{i \leftarrow 1}^{\psi^2 + n_0}$ and $\{z_i\}_{i \leftarrow 1}^{\psi^2 + n_0}$.

2: $X_{norm(i)} \leftarrow floor(\mathrm{mod}(abs(x_{i+n_0}) - floor(abs(x_{i+n_0})) \times 10^{14}, 16)) + 1$

3: $Y_{norm(i)} \leftarrow floor(\mathrm{mod}(abs(y_{i+n_0}) - floor(abs(y_{i+n_0})) \times 10^{14}, 16)) + 1$

4: $Z_{norm(i)} \leftarrow floor(\mathrm{mod}(abs(z_{i+n_0}) - floor(abs(z_{i+n_0})) \times 10^{14}, 16)) + 1$

5: $W_{norm(i)} \leftarrow \mathrm{mod}(X_{norm(i)} + Y_{norm(i)} + Z_{norm(i)}, 16) + 1$

6: $octagon\_centers \leftarrow zeros(\psi^2, 2)$

7: $radius\_steps \leftarrow zeros(\psi^2, 2)$

8: **for** $i \leftarrow 1 : \psi^2$ **do**

9: $octagon\_centers(i, :) \leftarrow [X_{norm(i)}, Y_{norm(i)}]$

10: $radius\_steps(i, :) \leftarrow [Z_{norm(i)}, W_{norm(i)}]$

11: **end for**

12: return $octagon\_centers,\ radius\_steps$

**Normalization:**
Lines (2-5) normalize the chaotic sequences into values between 1 and 16. $X_{norm}$, $Y_{norm}$, and $Z_{norm}$ represent normalized versions of the chaotic sequences $x$, $y$, and $z$. Besides, $W_{norm}$ combines the normalized values ($X_{norm}$, $Y_{norm}$, and $Z_{norm}$) to generate a fourth key stream.

**Initialize Arrays:**
Two arrays, *octagon_centers* and *radius_steps*, are being initialized to zeros. They will store the octagon center coordinates and radius/step values, respectively. Each array has $\psi^2$ rows and 2 columns (Line 6-7).

**Populate Arrays:**
The *octagon_centers* array is populated with $X_{norm}$ and $Y_{norm}$, representing the $x$ and $y$ coordinates of the centers. The *radius_steps* array is populated with $Z_{norm}$ (radius) and the $W_{norm}$ (step value) Lines (8-11).

**Return Results:**
The function returns the *octagon_centers* and *radius_steps* arrays, which are used in further cryptographic operations (Line 12). This key generation algorithm leverages chaotic sequences to create complex and unpredictable values, enhancing the security of the cryptographic system.

## 3.2 OCTA-SBox algorithm

Fig 10 shows the proposed methodology. Invoke the Algorithm (2) with the set of parameters *sbox*, *octagon_centers*, *radius_steps* and $\psi$ where $sbox = linspace(1, \psi, \psi)$.

The algorithm $OCTA-SBox$ is designed to generate a scrambled S-Box using octagonal patterns within an *sbox* array. Here's a breakdown of how the algorithm works:

**Reshape the sbox array:**
Reshape the *sbox* array to the size of $\sqrt{\psi} \times \sqrt{\psi}$ (Line 1).

**Loop Through Octagon Centers:**
The algorithm iterates through each element in the *octagon_centers* array (Line 2).
 *idx*: The index used to access the current octagon's center coordinates.

**Extract Octagon Parameters:**
For each octagon, the algorithm extracts its center from *octagon_centers* and its corresponding radius and steps from *radius_steps* (Lines 3-5).

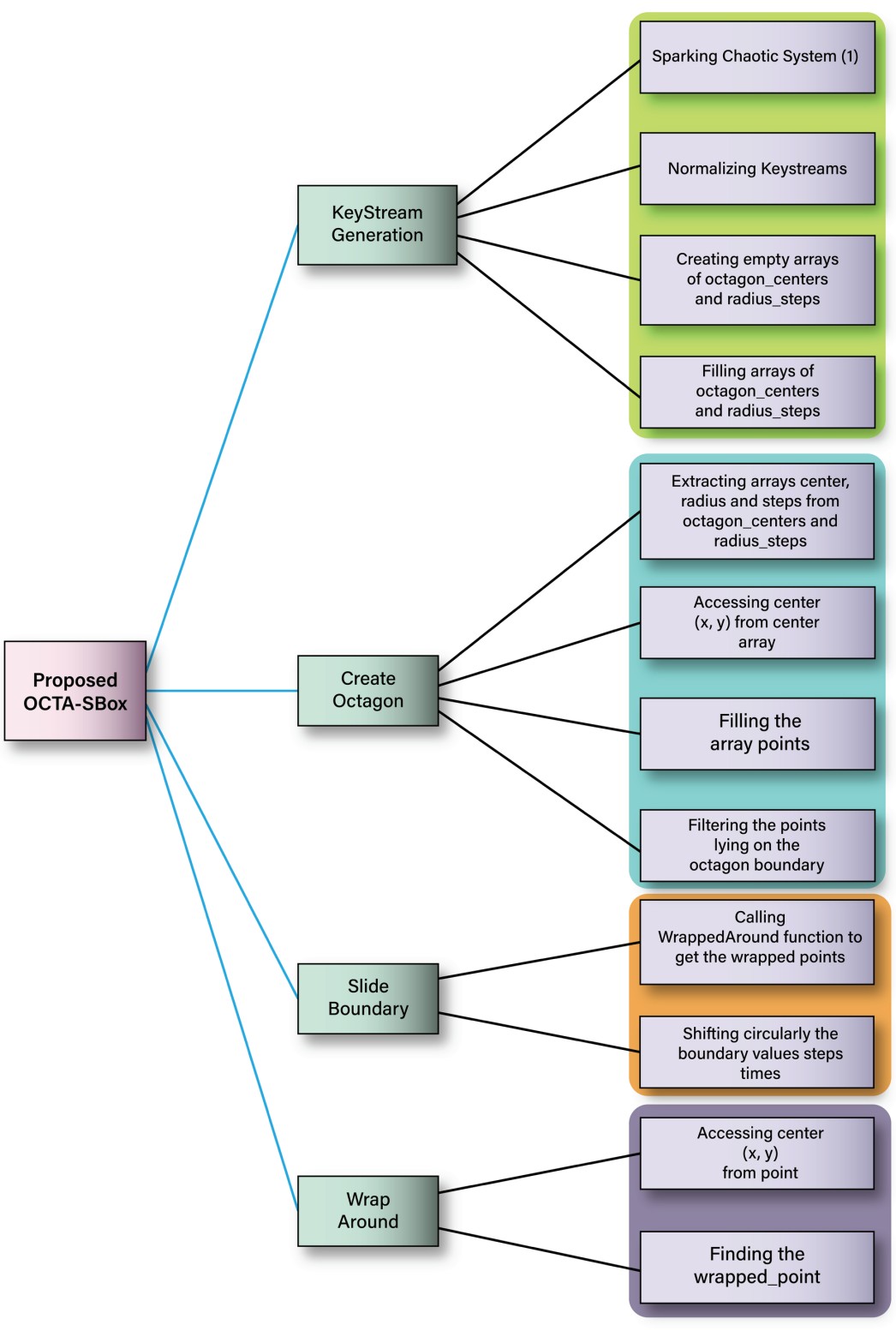

**Fig 10. Proposed methodology.**

**Algorithm 2.    *OCTA−SBox*.**

```
Input: sbox, octagon_centers, radius_steps, ψ
Output: sbox′
1:  sbox ← sbox.reshape(√ψ, √ψ)
2:  for idx ← 1 : length(octagon_centers) do
3:      center ← octagon_centers(idx, :)
4:      radius ← radius_steps(idx, 1)
5:      steps ← radius_steps(idx, 2)
6:      boundary_points ← CreateOctagon(center, radius, ψ)
7:      if ∼ isempty(boundary_points) then
8:          sbox ← SlideBoundary(sbox, boundary_points, steps)
9:      end if
10: end for
11: sbox′ ← sbox
12: return sbox′
```

**Create Octagon:**

The function *CreateOctagon* is called to calculate the boundary points *boundary_points* of the octagon based on its center and radius (Line 6).

**Check Boundary Points:**

If the calculated *boundary_points* is not empty, the algorithm proceeds to slide these points along the boundary of the octagon using the specified number of *steps* (Line 7).

**Slide Boundary:**

The function *SlideBoundary* shifts the elements located at the boundary points within the *sbox* by the specified number of *steps*. This step effectively scrambles the values on the boundary of the octagon (Line 8).

**End Loop:**

The loop continues until all octagons have been processed (Line 10).

**Return Scrambled S-Box**:

Finally, the scrambled S-Box *sbox′* is returned as the output of the algorithm (Line 12).

Here we will explain the *CreateOctagon* function.

Before explaining the function, we can see the Fig 11 in which the eight sides have been named. We will refer to them with their names during the explanation of the algorithms.

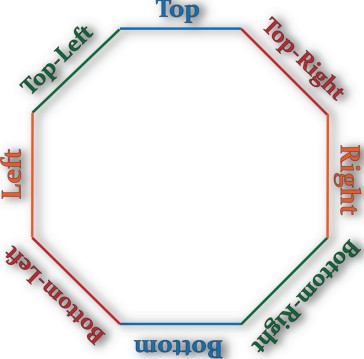

**Fig 11. The nomenclature being used for the eight sides of the octagon.**

**Extract $x$– and $y$–coordinates from the center:**

$x$ and $y$ are being assigned the values taken from the *center* array (Line 1-2). *points* is initialized as an empty list to be filled with data points later in the algorithm (Line 3).

**Loop Initialization:**

The loop iterates over a range from -*radius* to +*radius*. The variable $i$ represents an offset from the center point along the $x$ or $y$ direction (Line 4).

**Top and Bottom Sides:**

$x + i$: Adjusts the $x$-coordinate by $i$ while keeping the $y$-coordinate fixed at $y + radius$. This calculates the points along the top boundary of the octagon (Line 5). Similarly, $x + i$: Adjusts the $x$-coordinate by $i$ while keeping the $y$-coordinate fixed at $y$–*radius*. This calculates the points along the bottom boundary of the octagon (Line 6).

**Right and Left Sides:**

$y + i$: Adjusts the $y$-coordinate by $i$ while keeping the $x$-coordinate fixed at $x + radius$. This calculates the points along the right boundary of the octagon (Line 7).

$y + i$: Adjusts the $y$-coordinate by $i$ while keeping the $x$-coordinate fixed at $x$–*radius*. This calculates the points along the left boundary of the octagon (Line 8).

**Slant Sides (Top-Right, Bottom-Left, Top-Left, Bottom-Right):**

The condition $abs(i) < radius$ ensures that the slant sides are only calculated for points that are within the octagon's boundaries (Line 9).

**Top-Right and Bottom-Left Slant calculation:**

$x + radius - abs(i), y + i$: Adjusts the $x$-coordinate based on the distance $i$ from the center, ensuring the point lies on the slant side of the octagon (Line 10).

**Top-Left and Bottom-Right Slant calculation:**

$x - radius + abs(i), y + i$: Similar adjustment for the other diagonal side of the octagon (Line 11).

**Points Accumulation:**

In each iteration, the calculated points are accumulated into the *points* array, forming the boundary of the octagon (Lines 10-11).

**Filtering Points:**

The line 14 filters the *points* array, keeping only those points where both coordinates are within the bounds of a $\sqrt{\psi} \times \sqrt{\psi}$ matrix. Specifically, it ensures that the $x$-coordinate (first column) and the $y$-coordinate (second column) of each point are between 1 and $\sqrt{\psi}$.

**Return Points:**

Finally, line 15 returns the *points* array to the Algorithm *OCTA–SBox*.

Consider the line 7 of the Algorithm 2. If the *if* condition of the line 7 evaluates to be *false*, control shifts at the line 2 for the next iteration of the *for* loop. In case, the *if* condition at the line 7 evaluates to be *true*, *SlideBoundary* algorithm is invoked. Let's explain here this algorithm.

**Algorithm Overview:**

This algorithm is designed to slide (or shift) the integers along the boundary of an octagon-shaped region within the given sbox. This is done by a specified number of steps, creating a scrambled version of the sbox.

**Input Parameters**

*sbox*: The original matrix, e.g., $(\sqrt{\psi} \times \sqrt{\psi})$ where the octagon boundary is located.

*boundary_points*: A list of coordinates that define the points on the octagon boundary.

*steps*: The number of positions to shift the boundary values.

**Algorithm 3.** *CreateOctagon.*

```
Input: center, radius, ψ
Output: points
1: x ← center(1)
2: y ← center(2)
3: points ← [  ]
4: for i ← −radius : radius do
5:     points ← [points; x + i, y + radius]
6:     points ← [points; x + i, y − radius]
7:     points ← [points; x + radius, y + i]
8:     points ← [points; x − radius, y + i]
9:     if abs(i)<radius then
10:         points ← [points; x + radius − abs(i), y + i]
11:         points ← [points; x − radius + abs(i), y + i]
12:     end if
13: end for
14: points ← points(points(:, 1) ≥ 1  &  points(:, 1) ≤ √ψ  &  points(:, 2) ≥ 1  &  points(:, 2) ≤ √ψ)
15: return points
```

The variable *num_points* holds the number of boundary points (rows) in *boundary_points*. $size(boundary\_points, 1)$ returns the number of rows in the matrix, which corresponds to how many points are on the boundary (Line 1). The line 2 initializes the array *boundary_points* with zeros.

**Extracting Boundary Values from the sbox:**
Loop (Lines 3-6) iterates through each boundary point.
   *wrapped_pt*: This calls the *WrapAround* function, which ensures that if a point is out of the sbox's bounds, it wraps around to the other side. It effectively handles edge cases where the boundary might exceed the matrix limits. We will explain the *WrapAround* function shortly.
   *boundary_values(i)*: Retrieves the value from the sbox at the *wrapped_pt* location and stores it in the *boundary_values* array.
   *Perform Circular Shift*:
   *circshift*: This is the "meatiest" instruction performing the real "manufacturing" task. This function shifts the elements of *boundary_values* circularly by the amount specified in *step*. If *step* is positive, the shift is to the right; if negative, to the left. Moreover, it places the shifted values back into the *boundary_values* (Line 7).
**Loop**: Again iterates through each boundary point (Line 8).
 *wrapped_pt*: Ensures the point is within matrix bounds (Line 9).
**Assigning Values**: The shifted values are placed back into the *sbox* at their respective boundary positions (Line 10).

**Final Output**:
 The modified *sbox* is returned, with the boundary values shifted according to the specified *steps*. This creates a scrambled effect within the *sbox* based on the octagonal boundary. This algorithm is integral in creating a secure and randomized structure within the *sbox*, useful for encryption or other applications requiring complex data scrambling.
   The **WrapAround** algorithm is designed to adjust a point's coordinates to ensure they stay within the bounds of a grid, wrapping around if they exceed the boundaries.
**Extract Coordinates:** The *x* and *y* coordinates of the input point are extracted (Line 1-2).

**Wrap Around Calculation:**
The coordinates are adjusted using the modulus operation to ensure they stay within the grid's boundaries. The calculation $mod([x, y] − 1, [max\_x, max\_y]) + 1$ wraps the coordinates around if they exceed the grid limits (Line 3).

**Algorithm 4.** *SlideBoundary.*

```
Input: sbox, boundary_points, steps
Output: sbox
1:  num_points ← size(boundary_points, 1)
2:  boundary_values ← zeros(num_points, 1)
3:  for i ← 1 : num_points do
4:      wrapped_pt ← WrapAround(boundary_points(i, :), size(sbox, 1), size(sbox, 2))
5:      boundary_values(i) ← sbox(wrapped_pt(1), wrapped_pt(2))
6:  end for
7:  boundary_values ← circshift(boundary_values, steps)
8:  for i ← 1 : num_points do
9:      wrapped_pt ← WrapAround(boundary_points(i, :), size(sbox, 1), size(sbox, 2))
10:     sbox(wrapped_pt(1), wrapped_pt(2)) ← boundary_values(i)
11: end for
12: return sbox
```

**Return Wrapped Point:**

The adjusted (wrapped) coordinates are returned as *wrapped_point* to the calling algorithm.

**Algorithm 5.** *WrapAround.*

```
Input: point, max_x, max_y
Output: wrapped_point
1:  x ← point(1)
2:  y ← point(2)
3:  wrapped_point ← mod([x, y] − 1, [max_x, max_y]) + 1
4:  return wrapped_point
```

# 4 Simulation and performance analysis

As the Algorithm 1 was simulated by taking $\psi = 256$, we got the $16 \times 16$ S-Box (Table 1) drawn in hexadecimal numbers.

Merely developing cryptographic products is not enough; they must also adhere to the latest standards, benchmarks, and evaluation criteria established by analysts, security experts, and cryptographers. In this section, we will demonstrate the strength and resilience of the proposed S-Box using these benchmarks.

## 4.1 Bijectivity

Typically, an S-Box of size $Q \times Q$ is considered bijective if it contains $2^{Q-1}$ distinct values [27]. Moreover, these values span the range of $[0, 2^{Q-1}]$. It is clear that the S-Box presented in Table 1 meets this bijectivity condition.

## 4.2 Non-linearity

Potential attackers sometimes employ linear cryptanalysis to compromise security products [28]. To counter this threat, the designed S-Boxes must incorporate a high level of non-linearity. If a linear relationship exists between the plaintext and ciphertext, adversaries could exploit it to attack the S-Box. The inherent non-linearity of an $n$-bit Boolean function, such as $b(k)$, can be evaluated using the following mathematical equation (2) [29].

$$NL(b) = \frac{1}{2}\{2^n - max_{h \in \{0,1\}^n}|WS_b(h)|\} \tag{13}$$

**Table 1. Suggested S-Box.**

| i/j | 0 | 1 | 2 | 3 | 4 | 5 | 6 | 7 | 8 | 9 | A | B | C | D | E | F |
|---|---|---|---|---|---|---|---|---|---|---|---|---|---|---|---|---|
| 0 | 9D | 05 | 62 | 65 | B6 | AF | C1 | 8B | 77 | FB | 3F | 4E | 0B | 17 | 5A | B9 |
| 1 | E9 | 94 | 71 | 8F | FC | AB | DF | 9F | 5E | D8 | D6 | A0 | A5 | 83 | 98 | FD |
| 2 | B7 | 88 | F4 | 1B | 90 | 1E | 13 | 76 | DE | C5 | CC | 8D | 4B | 23 | 2E | 91 |
| 3 | 14 | 0C | 2B | AA | F8 | B1 | 43 | 02 | E8 | 79 | A2 | 92 | 41 | AC | 9C | B4 |
| 4 | 58 | 49 | 27 | 1F | 81 | EB | 82 | E0 | 42 | A7 | 60 | F1 | 3A | C0 | 30 | 1C |
| 5 | 6A | 54 | EE | 3E | 8E | 73 | 2A | 87 | 48 | 61 | 20 | EA | BB | 57 | 72 | 01 |
| 6 | E7 | 56 | 0A | FA | 0D | A8 | 37 | D2 | D9 | 2D | 74 | 46 | 29 | C9 | D4 | FF |
| 7 | A4 | 7F | E6 | F3 | 45 | CD | 85 | 68 | 6E | 35 | AD | 5F | 97 | CE | 51 | BD |
| 8 | 34 | CB | D5 | BE | 7B | DC | EC | 86 | 95 | CA | 39 | CF | 75 | 99 | 04 | 28 |
| 9 | 26 | 16 | 31 | 7A | 67 | 6C | 5B | 9B | 4C | B8 | AE | DD | C3 | 6F | B2 | 44 |
| A | B3 | 3C | 11 | F7 | 36 | E4 | 1A | E2 | 66 | C6 | 24 | F5 | D1 | B0 | 06 | ED |
| B | 8A | 03 | BA | 52 | A6 | 12 | 0F | DB | 50 | 6D | DA | 89 | 33 | A9 | 32 | 53 |
| C | D7 | C8 | 15 | 93 | B5 | C4 | 38 | 4A | 6B | BF | 55 | F2 | 10 | F0 | C2 | E1 |
| D | 4F | 3B | 7C | 18 | 4D | BC | E5 | 1D | 40 | 9A | C7 | 19 | E3 | D0 | 7E | 78 |
| E | FE | 2C | 21 | 59 | 9E | 7D | 96 | 2F | 25 | 5D | F6 | 69 | 8C | F9 | EF | A1 |
| F | 80 | D3 | A3 | 09 | 5C | 08 | 64 | 22 | 0E | 10 | 84 | 70 | 3D | 07 | 47 | 63 |

In this equation, $WS_b(h)$ represents the Walsh spectrum of the function $b$. Additionally, the non-linearity of the $n$-bit Boolean function $b(k)$ can be determined using the following mathematical expression.

$$WS_{b(h)} = \sum_{x \in \{0,1\}^n} (-1)^{b(x) \oplus h.x} \tag{14}$$

The equation $h \in \{0,1\}^n$ is provided, where the dot product between $h$ and $x$ is represented by $h.x$. This dot product can be calculated using the following method.

$$h.x = (h_1 \oplus x_1) + ... + (h_n \oplus x_n) \tag{15}$$

The non-linearity values for the proposed S-Box are 107, 102, 103, 105, 107, 108, 109, and 104. The minimum, maximum, and average values are 102, 109, and 105.625, respectively. The non-linearity for each of the eight Boolean functions is detailed in Table 2.

## 4.3 Strict-avalanche criterion (SAC)

To satisfy the strict avalanche criterion (SAC), changing a single input bit $n$ should result in a 50% chance of altering the corresponding output bit $m$ [29]. In other words, the S-Box exhibits sufficient unpredictability and chaotic behavior if its SAC value is close to 0.5. The SAC values calculated for the proposed S-Box are presented in Table 3, also known as the dependency matrix. The average SAC value of 0.508391 for the proposed S-Box meets the established standard.

**Table 2. Suggested S-Box nonlinearity values.**

| Boolean function | $f_1$ | $f_2$ | $f_3$ | $f_4$ | $f_5$ | $f_6$ | $f_7$ | $f_8$ |
|---|---|---|---|---|---|---|---|---|
| Non-linearity values | 107 | 102 | 103 | 105 | 107 | 108 | **109** | 104 |

**Table 3. Suggested S-Box SAC results.**

| i/j | 1 | 2 | 3 | 4 | 5 | 6 | 7 | 8 |
|---|---|---|---|---|---|---|---|---|
| 1 | 0.5273 | 0.5498 | 0.5213 | 0.5176 | 0.4897 | 0.5098 | 0.5178 | 0.4634 |
| 2 | 0.5287 | 0.4789 | 0.5039 | 0.4907 | 0.4852 | 0.5178 | 0.4890 | 0.5264 |
| 3 | 0.5498 | 0.5287 | 0.5388 | 0.5299 | 0.5311 | 0.4972 | 0.5123 | 0.5098 |
| 4 | 0.5193 | 0.5240 | 0.5095 | 0.5297 | 0.4621 | 0.4908 | 0.5033 | 0.5184 |
| 5 | 0.4954 | 0.4987 | 0.4783 | 0.5024 | 0.5124 | 0.5206 | 0.4827 | 0.5214 |
| 6 | 0.5264 | 0.4756 | 0.4806 | 0.5274 | 0.5133 | 0.5221 | 0.5150 | 0.5244 |
| 7 | 0.5181 | 0.4623 | 0.4900 | 0.5211 | 0.5113 | 0.4980 | 0.5257 | 0.5222 |
| 8 | 0.5103 | 0.4912 | 0.5023 | 0.4813 | 0.5284 | 0.5124 | 0.4914 | 0.5023 |

## 4.4 Bit-independence criterion (BIC)

Cryptographers use this additional criterion to evaluate the robustness of their S-Box. According to this benchmark, an S-Box is deemed effective at isolating output bits if a change in a single input bit, such as $q$, results in changes in the output bits $r$ and $s$ independently [29]. To achieve this, the Boolean functions that form the S-Box must satisfy specific non-linearity conditions. The performance of the S-Box, denoted as $T$, is assessed by calculating the difference $(T_a[p] \oplus T_b[q]) - (T_a[p] \oplus T_b[p])$ over the entire range of input values $p$ from 0 to 255. This measure, known as the BIC-SAC performance, evaluates how well the S-Box maintains the BIC property. For this evaluation, $p$ and $q$ should differ by exactly one bit. The average BIC-SAC values, which reflect the efficacy of the S-Box, should be close to 0.5 for optimal performance. Tables 4 and 5 show the criteria used for assessing SAC and non-linearity in the component Boolean functions of the proposed S-Box. The mean non-linearity and SAC values for the proposed S-Box are 103.267857 and 0.5090107, respectively.

According to Carlisle and Stafford [30], an S-Box is considered to have the BIC property if it meets both non-linearity and SAC requirements. The values of 103.267857 and 0.5090107 for the proposed S-Box indicate a notably weak linear relationship among the output bits, affirming that the proposed S-Box indeed exhibits the BIC property.

## 4.5 Linear-probability (LP)

The concept of linear probability is essential for assessing the inherent relationship between an S-Box's input and output. A lower linear probability (LP) value indicates a robust S-Box. In the case of the suggested S-Box, the equation below yields a maximum LP value of 0.1224:

$$LP = \max_{a_z, b_z \neq 0} \left| \frac{\#\{z \in N | z.a_z = T(z).b_z\}}{2^n} - \frac{1}{2} \right|$$

**Table 4. Suggested S-Box's BIC non-linearity results.**

| i/j | 1 | 2 | 3 | 4 | 5 | 6 | 7 | 8 |
|---|---|---|---|---|---|---|---|---|
| 1 | - | 105 | 103 | 100 | 102 | 102 | 109 | 104 |
| 2 | 107 | - | 102 | 107 | 100 | 102 | 100 | 107 |
| 3 | 99 | 104 | - | 107 | 103 | 100 | 107 | 102 |
| 4 | 104 | 102 | 108 | - | 106 | 102 | 105 | 103 |
| 5 | 101 | 104 | 99 | 104 | - | 109 | 101 | 102 |
| 6 | 101 | 100 | 100 | 104 | 101 | - | 106 | 102 |
| 7 | 107 | 106 | 103 | 102 | 104 | 100 | - | 101 |
| 8 | 104 | 102 | 106 | 102 | 107 | 101 | 102 | - |

**Table 5. Suggested S-Box BIC-SAC results.**

| i/j | 1 | 2 | 3 | 4 | 5 | 6 | 7 | 8 |
|---|---|---|---|---|---|---|---|---|
| 1 | - | 0.4855 | 0.5255 | 0.5099 | 0.5099 | 0.5188 | 0.5047 | 0.5084 |
| 2 | 0.5044 | - | 0.5033 | 0.5032 | 0.5173 | 0.4974 | 0.5035 | 0.5026 |
| 3 | 0.5155 | 0.5016 | - | 0.4877 | 0.5056 | 0.5288 | 0.5299 | 0.5133 |
| 4 | 0.5432 | 0.4849 | 0.5374 | - | 0.5133 | 0.5241 | 0.5289 | 0.4722 |
| 5 | 0.5173 | 0.5233 | 0.5123 | 0.5204 | - | 0.5049 | 0.5099 | 0.5063 |
| 6 | 0.5059 | 0.5113 | 0.5374 | 0.4825 | 0.5104 | - | 0.4968 | 0.5023 |
| 7 | 0.5412 | 0.4917 | 0.4916 | 0.4977 | 0.5098 | 0.5114 | - | 0.5234 |
| 8 | 0.4983 | 0.4912 | 0.5345 | 0.5034 | 0.5056 | 0.4815 | 0.5015 | - |

In this equation,

- $T$ represents the S-Box.
- $a_z$ and $b_z$ are input and output masks, respectively.
- $N$ encompasses integers from 0 to 255.

This LP value suggests that the S-Box is sufficiently resilient against linear cryptanalysis attempts.

## 4.6 Differential probability (DP)

In differential cryptanalysis, the goal is to recover the initial plaintext by analyzing the differences between pairs of ciphertexts and their corresponding plaintexts. Adversaries and cryptanalysts exploit these discrepancies to gain access to the secret key. For a robust S-Box, this statistical measure should have a relatively low value.

The notion of differential probability (DP) quantifies the likelihood of a specific output differential given an input differential. It can be calculated using the following equation:

$$DP = \max_{\triangle_z \neq 0, \triangle_y} \left| \frac{\#\{z \in N | T_{(z)} \oplus T_{(z \oplus \triangle z)} = \triangle y\}}{2^n} \right|$$

In this equation,

- $T$ represents the S-Box.
- $\triangle y$ and $\triangle z$ denote the output and input differentials, respectively.
- $N$ covers integers from 0 to 255.

According to the evidence in Table 6, the suggested S-Box yields a differential probability of $10/256 = 0.0391$. This result indicates strong defense against various differential cryptanalysis attacks.

## 4.7 Fixed and reverse fixed points analysis

Many S-Boxes have been published which contain the fixed and reverse fixed points vulnerabilities [45]. Fixed points occur in the S-Boxes if the elements contained by the boxes are mapped to themselves. In this way, the given input becomes equal to the output which can be exploited by the potential hackers. Besides, the phenomenon of reverse fixed point happens if the particular element of S-Box is mapped to its corresponding binary complement. For instance, if 11001010 (202 in decimal) is placed in some input and its output is its binary

**Table 6. Suggested S-Box DP table.**

| i/j | 0 | 1 | 2 | 3 | 4 | 5 | 6 | 7 | 8 | 9 | A | B | C | D | E | F |
|---|---|---|---|---|---|---|---|---|---|---|---|---|---|---|---|---|
| 0 | 8 | 6 | 4 | 6 | 6 | 8 | 6 | 6 | 6 | 6 | 8 | 6 | 8 | 6 | 8 | 6 |
| 1 | 6 | 6 | 6 | 6 | 6 | 8 | 6 | 8 | 6 | 8 | 6 | 6 | 8 | 6 | 6 | 8 |
| 2 | 6 | 6 | 6 | 6 | **10** | 8 | 6 | 6 | 6 | 6 | 6 | 6 | 6 | 6 | 6 | 6 |
| 3 | 6 | 6 | 6 | 8 | 6 | 6 | 6 | 6 | 8 | 6 | 6 | 6 | 8 | 6 | 6 | 6 |
| 4 | 6 | 8 | 6 | 6 | 8 | 6 | 6 | 6 | 6 | 6 | 6 | 8 | 6 | 6 | 6 | 8 |
| 5 | 6 | 6 | 8 | 6 | 6 | 6 | 6 | 6 | 6 | 6 | 6 | 6 | 8 | 6 | 6 | 6 |
| 6 | 6 | 6 | 8 | 6 | 6 | 6 | 6 | 6 | 6 | 6 | 6 | 6 | 6 | 6 | 6 | 6 |
| 7 | 6 | 8 | 8 | 6 | 6 | 6 | 8 | 8 | 8 | 8 | 6 | 8 | 8 | **10** | 6 | 8 |
| 8 | 6 | 8 | 6 | 4 | 6 | 8 | 6 | 6 | 8 | 6 | 6 | 8 | 6 | 6 | 8 | 6 |
| 9 | 6 | 6 | 8 | 6 | 6 | 6 | 4 | 6 | 8 | 6 | 6 | 6 | 8 | 6 | 6 | 6 |
| A | 6 | 8 | 6 | 6 | 6 | 6 | 8 | 6 | 8 | 6 | 8 | 6 | 8 | 6 | 8 | 6 |
| B | 4 | 6 | 6 | 6 | 6 | 8 | 6 | 6 | 6 | 6 | 8 | 6 | 8 | 6 | 8 | 8 |
| C | 8 | 6 | 6 | 6 | 8 | 6 | 6 | 6 | 6 | 8 | 8 | 6 | 6 | 8 | 6 | 8 |
| D | 8 | 6 | **10** | 8 | 6 | 6 | 6 | 8 | 6 | 6 | 6 | 6 | 8 | 6 | 6 | 6 |
| E | 6 | 6 | 6 | 8 | 6 | 6 | 6 | 6 | 6 | 6 | 8 | 6 | 6 | 6 | 6 | 6 |
| F | 6 | 8 | 6 | 6 | 8 | 6 | 8 | 6 | 6 | 6 | 6 | 6 | 8 | 4 | 6 | - |

complement i.e., 00110101 (53 in decimal). Cryptographers must check their S-Boxes while designing them that there should not be any fixed or reverse fixed points in the S-Boxes. In case, they occur, the adversaries and attackers may breach the underlying security of the systems employing the vulnerable S-Boxes [43,44]. Table 7 shows the number of fixed and reverse fixed points of the proposed study which are 0 and 0 respectively. Besides, the proposed work has also been compared with other state of the art researches. One can see that the suggested work beats the studies [45–54] as far as the metric of fixed and reverse fixed point is concerned.

## 4.8 Attacks analyses: Chosen plaintext, known plaintext & ciphertext only

In order to demonstrate the defiance of the suggested S-Box from the varied cryptographic attack, we have chosen the Baboon and the two special images. These two special images are Black and White. Figs 12–20 show the encryption and decryption of these chosen images. The suggested S-Box carried out a very seminal role while encrypting and decrypting the chosen plaintext images.

**Table 7. Fixed and reverse fixed points analysis.**

| S-Box algorithm | Fixed points | Reverse fixed points |
|---|---|---|
| Ref. [45] | 4 | 1 |
| Ref. [46] | 2 | 1 |
| Ref. [47] | 2 | 1 |
| Ref. [48] | 2 | 1 |
| Ref. [49] | 2 | 5 |
| Ref. [50] | 1 | 3 |
| Ref. [51] | 1 | 2 |
| Ref. [52] | 1 | 1 |
| Ref. [53] | 1 | 2 |
| Ref. [54] | 1 | 4 |
| Ref. [55] | 0 | 0 |
| Ref. [56] | 0 | 0 |
| Proposed | 0 | 0 |

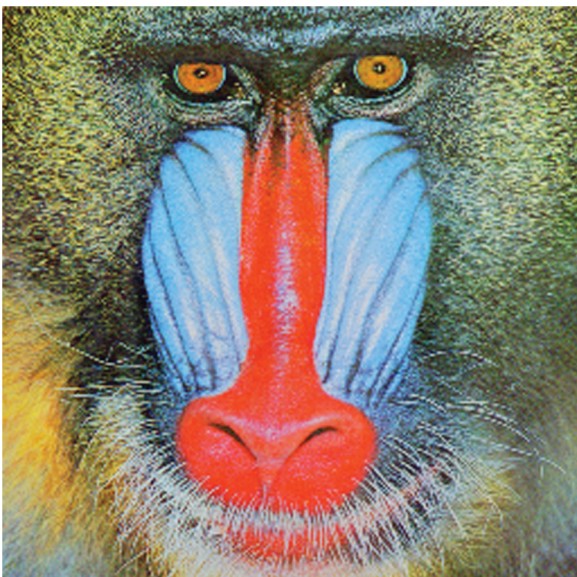

**Fig 12. Baboon plain image.**

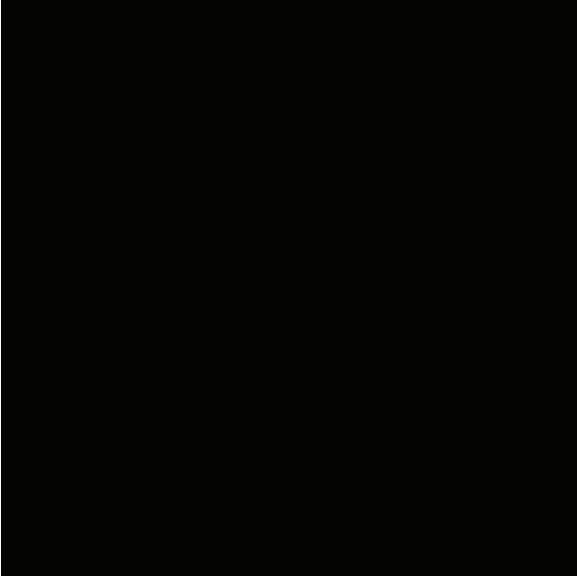

**Fig 13. Black image.**

In the field of cryptanalysis, hackers often employ plaintext and ciphertext attacks to compromise cryptosystems [57]. In a ciphertext-only attack, the hacker has access to one or more ciphertexts. In a known plaintext attack, the hacker possesses one or more plaintext-ciphertext pairs. In a chosen plaintext attack, the hacker temporarily gains access to the encryption mechanism, allowing them to generate as many ciphertexts as desired for chosen plaintexts. If we demonstrate that the proposed image cipher is resistant to chosen plaintext attacks, it

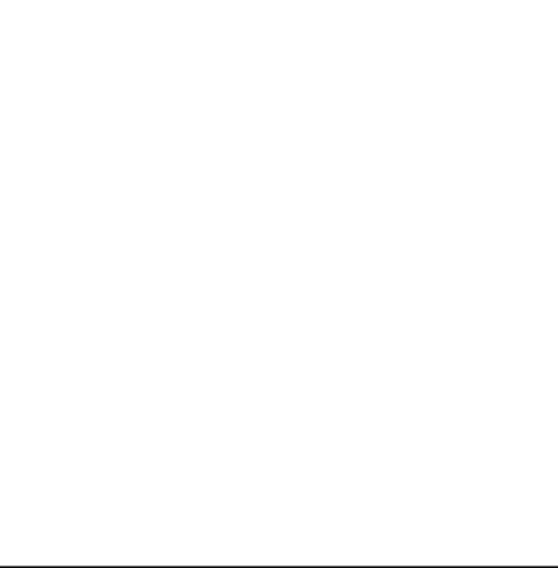

**Fig 14. White image.**

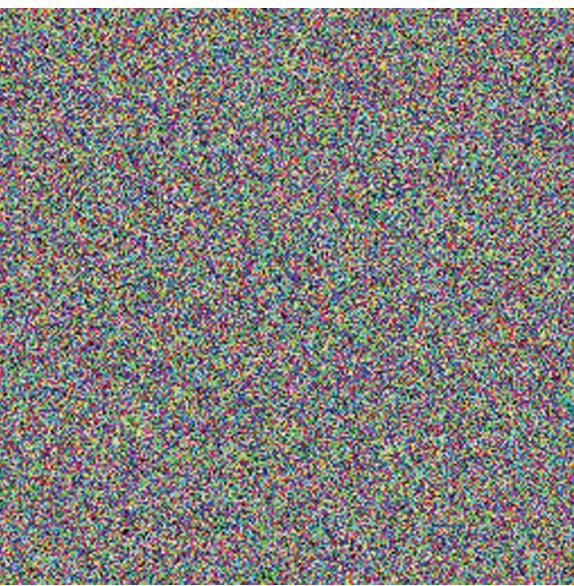

**Fig 15. Baboon encrypted image.**

inherently ensures immunity against known plaintext and ciphertext-only attacks, as these are considered subsets of the chosen plaintext attack.

In a chosen plaintext attack, the hacker temporarily gains access to the encryption algorithm and may select a specific image, such as a Black image (Fig 21), for encryption to deduce the secret key used in the process. After encrypting the Black image (Fig 22), the chaotic data utilized in the encryption algorithm (the secret key) is exposed. Subsequently, the hacker may select an arbitrary plain image, such as Baboon, and attempt to encrypt it

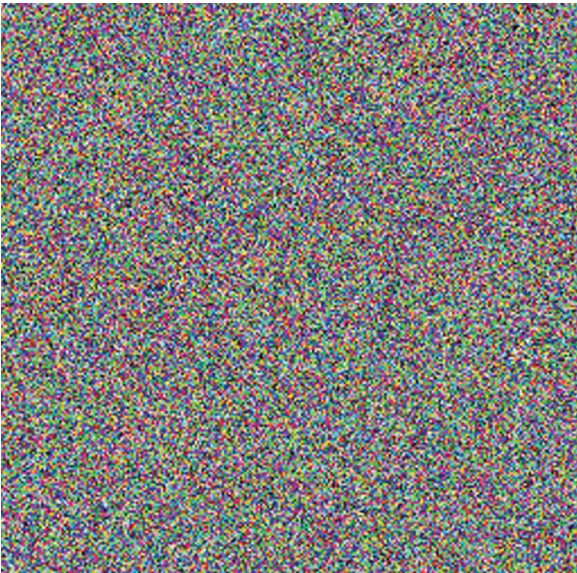

**Fig 16. Black encrypted image.**

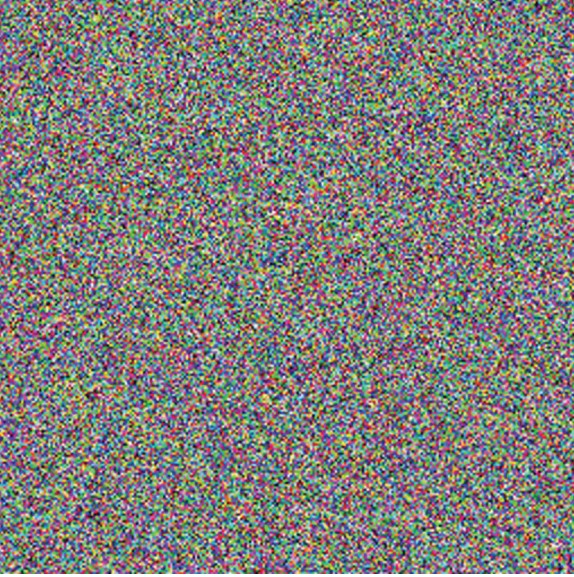

**Fig 17. White encrypted image.**

(Fig 23) by launching a known plaintext attack using the secret key obtained from the Black image encryption. However, to their frustration, the encrypted Baboon image cannot be successfully decrypted using this key (Fig 24). A similar process is repeated using another specific image, a White image, as shown in Figs 25–28. This process effectively demonstrates the cipher's resistance to chosen plaintext, known plaintext, and ciphertext-only attacks.

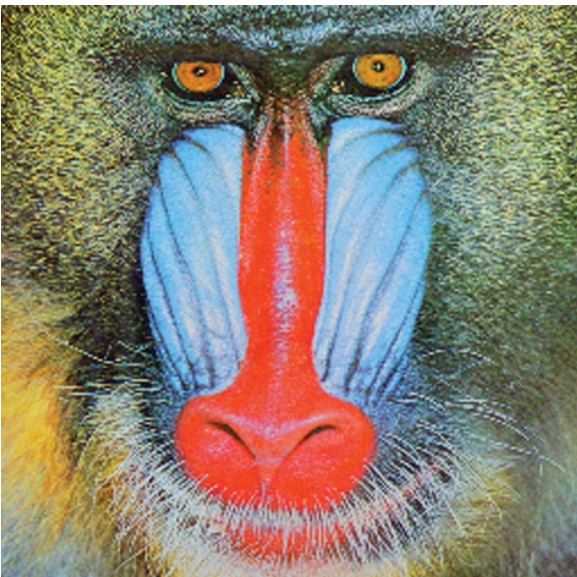

**Fig 18. Baboon decrypted image.**

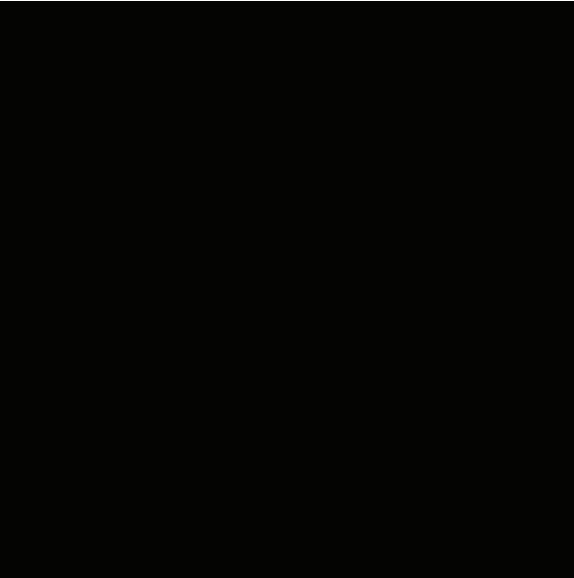

**Fig 19. Black decrypted image.**

## 5 Discussion

In today's high-tech world, where digital security threats are increasingly becoming sophisticated and pervasive, S-Boxes play a critical role in safeguarding sensitive information. These substitution boxes are integral to encryption algorithms, providing a crucial layer of non-linearity and confusion that transforms plaintext into secure ciphertext. Their ability

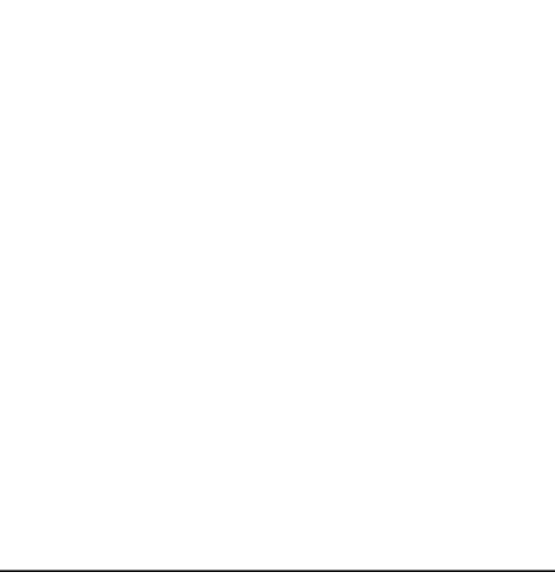

**Fig 20. White decrypted image.**

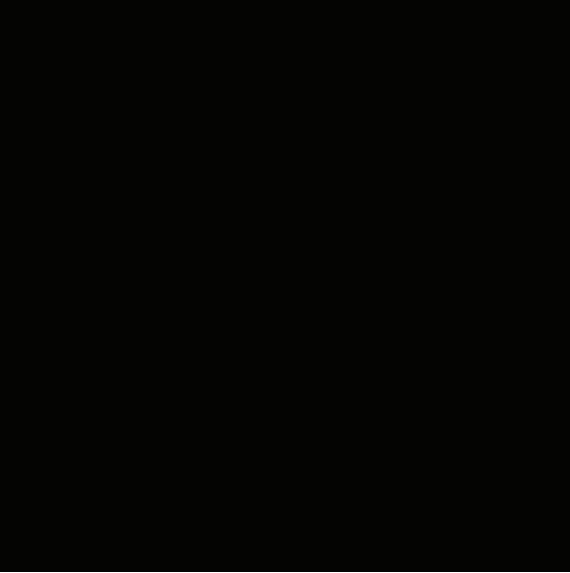

**Fig 21. Black plaintext image.**

to obscure patterns and resist cryptanalysis is essential for protecting data from unauthorized access and ensuring the integrity of secure communications. As technology evolves and cyber-attacks become more advanced, the necessity for robust, well-designed S-Boxes grows, highlighting their fundamental importance in maintaining security and privacy in a digitally interconnected world.

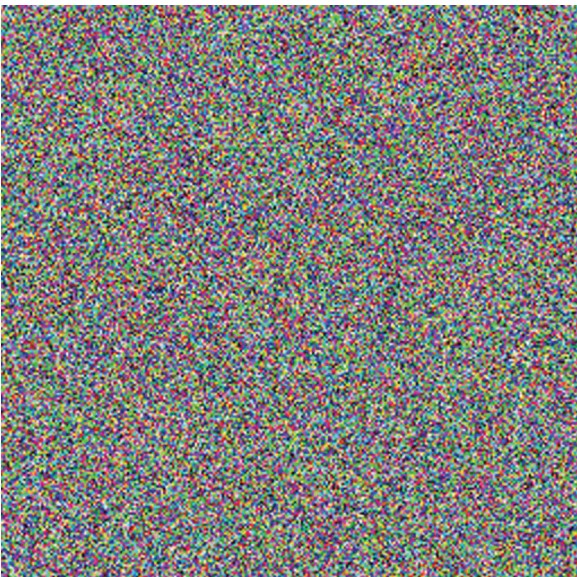

**Fig 22. Black ciphertext image.**

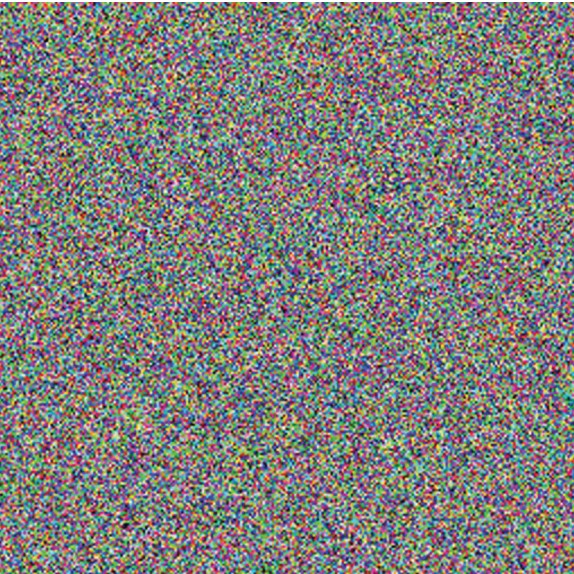

**Fig 23. Baboon cipher image.**

The integration of octagonal geometry in the current work adds another layer of sophistication. Octagonal geometry introduces a higher degree of symmetry and structural complexity, which enhances the S-Box's ability to distribute and mix data more thoroughly. This geometric approach enables the S-Box to exploit the spatial properties of the octagon, leading to more robust and non-linear transformations. The geometric properties ensure that each element of the S-Box undergoes a unique transformation, contributing to the overall strength and unpredictability of the cryptographic system.

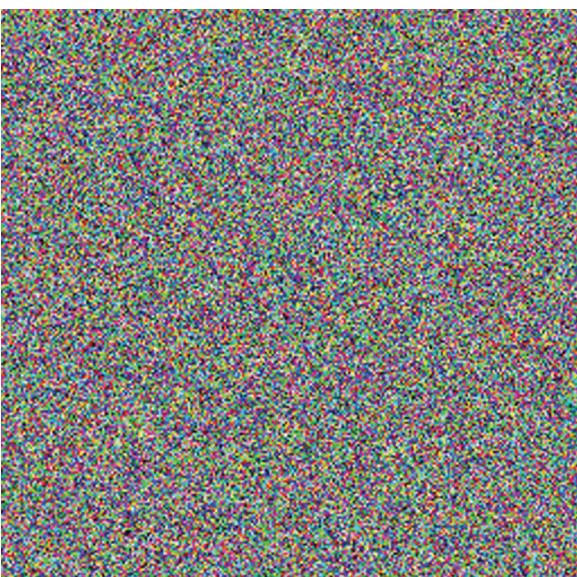

**Fig 24. Decrypted Baboon image with possible secret key from the Black image.**

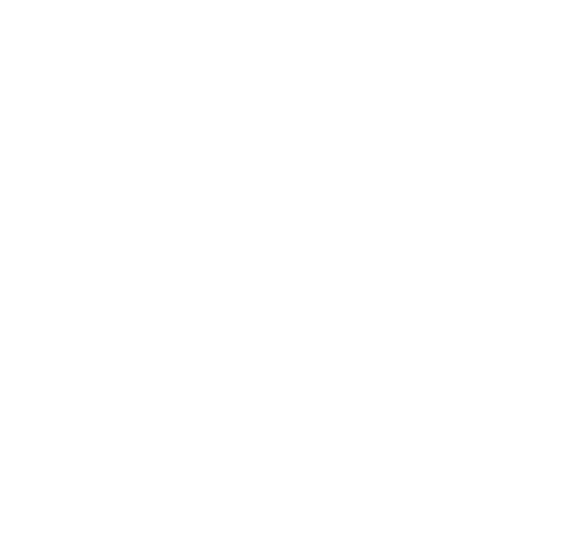

**Fig 25. White ciphertext image.**

The results obtained from this approach are promising as shown in the Table 8. Empirical evaluations indicate that the proposed S-Box exhibits superior resistance to various forms of cryptanalysis compared to traditional S-Boxes. The statistical analyses reveal that the S-Box maintains a high level of confusion and diffusion, crucial for the security of cryptographic algorithms. Additionally, the S-Box's performance in terms of avalanche effect and uniformity of output further supports its effectiveness.

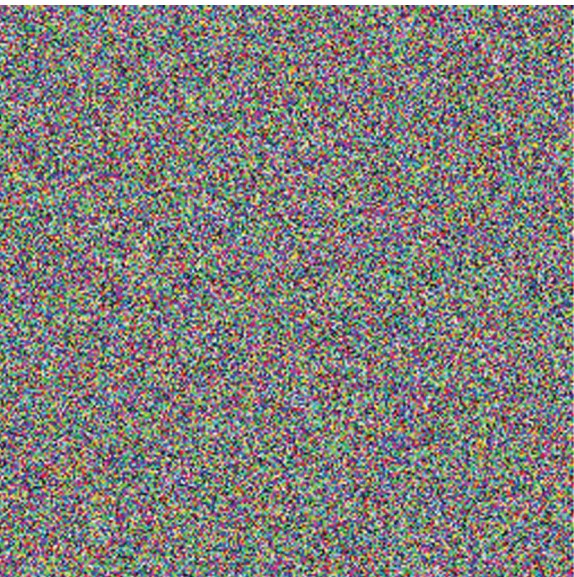

**Fig 26. White ciphertext image.**

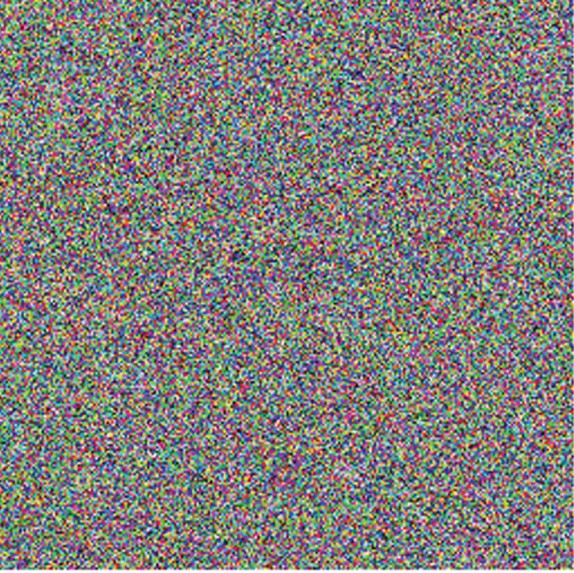

**Fig 27. Baboon ciphertext image.**

In summary, the development of the S-Box through the integration of chaos theory and octagonal geometry marks a noteworthy contribution to cryptographic technology. The enhanced security features and promising results underscore the value of interdisciplinary approaches in the advancement of cryptographic techniques. This work not only addresses current security challenges but also paves the way for future innovations in the field.

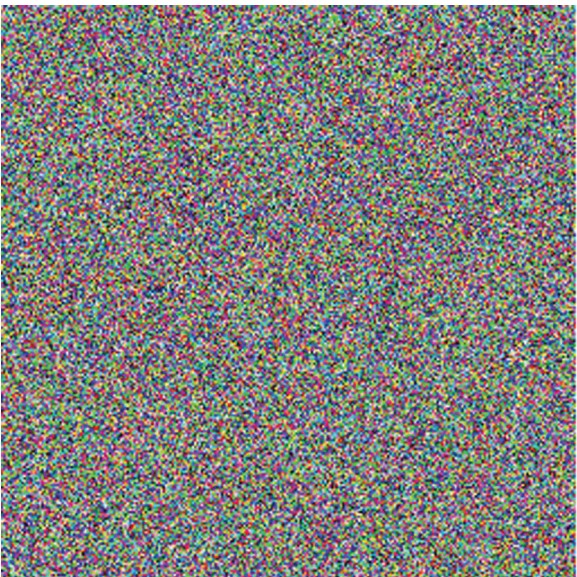

**Fig 28. Decrypted Baboon image with possible secret key from the White image.**

**Table 8. Comparative analysis of suggested S-Box with other published ones.**

| Work | Major theories of the scheme | NL | BIC-NL | SAC | BIC-SAC | LP | DP |
|---|---|---|---|---|---|---|---|
| Ref. [1] | Rook and 5D chaotic system | 106.125 | 103.17 | 0.5077 | 0.5061 | 0.1328 | 0.0469 |
| Ref. [31] | logistic-sine map | 105.25 | 103.8 | 0.4956 | 0.4996 | 0.1562 | 0.0391 |
| Ref. [32] | quantum-inspired QW | 106.00 | 103.9 | 0.4958 | 0.5023 | 0.1250 | 0.0313 |
| Ref. [33] | Mackey–Glass equation | 104.00 | 102.9 | 0.5000 | 0.4980 | 0.1328 | 0.0391 |
| Ref. [4] | Enhanced logistic map and Latin square | 105.25 | 103.2 | 0.5351 | 0.5000 | – | 0.0391 |
| Ref. [35] | Quantum-inspired QW and the customized PSO | 107.00 | 103.0 | 0.5044 | 0.5066 | 0.1172 | 0.0313 |
| Ref. [34] | Gingerbreadman chaotic system | 102.00 | 102.9 | 0.5178 | 0.4999 | 0.1250 | 0.0313 |
| Ref. [36] | Jaya optimization algorithm | 106.25 | 103.64 | 0.5009 | 0.4996 | 0.1171 | 0.0391 |
| Ref. [37] | 3D chaotic map | 106.00 | 104.2 | 0.4993 | 0.5030 | 0.1250 | 0.0391 |
| Ref. [38] | Teaching-learning-based optimization | 106.50 | 104.6 | 0.4995 | 0.4983 | 0.1172 | 0.0391 |
| Ref. [39] | Delannoy numbers and chaotic system | 105.875 | 103.179 | 0.5084 | 0.5087 | 0.1288 | 0.0391 |
| Ref. [40] | Chaotic map based on trigonometric functions | 110 | 102.78 | 0.5001 | - | 0.1328 | 0.03906 |
| Ref. [41] | Non-permutation binomial power functions | 108 | 106.32 | 0.5001 | - | 0.1545 | 0.015625 |
| Ref. [42] | Mordell Elliptic Curves over Galois Field | 112 | 106.32 | 0.5032 | 0.5059 | 0.0625 | 0.0156 |
| Ref. [27] | Chaotic map | 104.25 | 104 | 0.5029 | 0.5059 | 0.127 | 0.0391 |
| Proposed | Octogonal geometry and chaos | 105.625 | 103.267857 | 0.508391 | 0.5090107 | 0.1224 | 0.0391 |

## 6 Conclusion

By marrying the theory of chaos and geometrical figure octagon, this research endeavor has written a yet another S-Box to safeguard the digital assets. Firstly, a "raw" 2D S-Box (matrix) was created with 1 to 256 integers. To convert this matrix into a sophisticated S-Box, the instrument of octagon carried out a seminal role. Octagons of different radii and of different centers were hypothetically generated within this matrix to scramble the integers. The integers lying on the boundary of octagon were shifted circularly both clockwise and anti-clockwise. This operation has been repeated a lot of times to embed the security effects in the required S-Box. In case, some portion of the octagon goes past its any edge, the numbers

have been wrapped out—an other layer of complexity. The random data has been generated with the help of the Lorenz chaotic system. Both the simulation and security analysis rendered very satisfactory results. Given these findings, we assert that the proposed S-Box can be used in various domains of cryptographic applications like health, finance, business, e-government etc.

## 7 Future work

Building upon the foundational work of this research, future efforts will concentrate on the practical deployment of the proposed S-Box for enhancing digital image scrambling techniques. The primary goal is to integrate the S-Box into advanced image encryption algorithms and evaluate its performance in real-world scenarios. This involves assessing the S-Box's effectiveness in various image types, such as grayscale and color images, and measuring its impact on encryption quality and computational efficiency.

In addition to image encryption, future research will investigate the S-Box's potential in other critical areas of digital security. One promising avenue is its application in secure video transmission, where maintaining the integrity and confidentiality of video data is crucial. The S-Box's performance will be tested in video encryption frameworks to determine its suitability for protecting streaming and stored video content.

Another area of exploration will be the use of the S-Box in digital watermarking. Watermarking is essential for copyright protection and content verification, and the proposed S-Box could enhance watermarking techniques by improving their robustness against tampering and unauthorized access. Furthermore, the research will aim to optimize the S-Box for computational efficiency, ensuring that it performs well in real-time applications. This involves refining the algorithm to reduce processing time and resource usage, making it practical for use in environments with stringent performance requirements.

By exploring these diverse applications and focusing on optimization, the future work aims to demonstrate the versatility of the proposed S-Box and its potential to advance cryptographic techniques contributing to broader areas of digital security.

## Author contributions

**Conceptualization:** Abdulbasid Banga.

**Formal analysis:** Naif Al Mudawi.

**Investigation:** Nadeem Iqbal.

**Methodology:** Abdulbasid Banga, Nisreen Innab, Nadeem Iqbal, Hossam Diab.

**Project administration:** Naif Al Mudawi.

**Software:** Yasir Mahmood, Nisreen Innab.

**Validation:** Yasir Mahmood, Naif Al Mudawi, Hossam Diab.

**Writing – original draft:** Abdulbasid Banga, Yasir Mahmood, Nisreen Innab.

**Writing – review & editing:** Naif Al Mudawi, Nadeem Iqbal, Hossam Diab.

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
