## [Decision Letter · Decision Letter 0]

PONE-D-24-40373Where Octagonal Geometry Meets Chaos: A New S-Box for Advanced Cryptographic SystemsPLOS ONE

Dear Dr. Mahmood,

Thank you for submitting your manuscript to PLOS ONE. After careful consideration, we feel that it has merit but does not fully meet PLOS ONE’s publication criteria as it currently stands. Therefore, we invite you to submit a revised version of the manuscript that addresses the points raised during the review process.

We look forward to receiving your revised manuscript.

Kind regards,

Haider TH. Salim ALRikabi, Ph.D.

Academic Editor

PLOS ONE

2. Please note that PLOS ONE has spec6ific guidelines on code sharing for submissions in which author-generated code underpins the findings in the manuscript. In these cases, all author-generated code must be made available without restrictions upon publication of the work. Please review our guidelines at https://journals.plos.org/plosone/s/materials-and-software-sharing#loc-sharing-code and ensure that your code is shared in a way that follows best practice and facilitates reproducibility and reuse.

“This research is fully funded by United Arab Emirates University (UAEU) under the UAEU Start-Up Grant No. G00004635 and Fund No. 12T051.”

Reviewers' comments:

Reviewer's Responses to Questions

**Comments to the Author**

1. Is the manuscript technically sound, and do the data support the conclusions?

Reviewer #1: Yes

Reviewer #2: Yes

2. Has the statistical analysis been performed appropriately and rigorously? 

Reviewer #1: Yes

Reviewer #2: Yes

3. Have the authors made all data underlying the findings in their manuscript fully available?

Reviewer #1: Yes

Reviewer #2: Yes

4. Is the manuscript presented in an intelligible fashion and written in standard English?

Reviewer #1: Yes

Reviewer #2: Yes

5. Review Comments to the Author

Reviewer #1: In simulation and performance analysis, authors are required to add fixed points, conduct reverse fixed-point analysis for further improvements, and check validity against various cryptographic attacks.

Reviewer #2: The paper has a new design of S-Box based on chaos algorithm and Octagonal Geometry. The authors discuss the design and analyze the results. I think that there is a scientific contribution to be publish after correcting the following:

1. Overall text need to justify.

2. Section 5 (Discussion), I think this section may be merge with the section 4 (Optional).

3. Check all abbreviations.

4. The paper should be submitted carefully to the template of the journal.

6. PLOS authors have the option to publish the peer review history of their article (what does this mean?). If published, this will include your full peer review and any attached files.

Reviewer #1: No

Reviewer #2: No

---

## [Author Response · Author response to Decision Letter 1]

21 Jan 2025

Reviewer #1:

In simulation and performance analysis, authors are required to add fixed points, conduct reverse fixed-point analysis for further improvements, and check validity against various cryptographic attacks.

Answer: Thank you for your esteemed review. Fixed points and reverse fixed-point analyses have been carried out (Kindly see the subsection 4.7). Besides, cryptographic attacks analyses have also been made (Kindly see the subsection 4.8).

Reviewer #2:

The paper has a new design of S-Box based on chaos algorithm and Octagonal Geometry. The authors discuss the design and analyze the results. I think that there is a scientific contribution to be publish after correcting the following:

1. Overall text need to justify.

Answer: Thank you for your esteemed review. We have used the Latex template of the journal. So, this is the default behavior of the journal.

2. Section 5 (Discussion), I think this section may be merge with the section 4 (Optional).

Answer: Thank you for your valuable suggestion regarding merging Section 5 (Discussion) with Section 4 (Analysis). However, we believe that keeping these sections separate enhances the clarity and structure of the article. Section 4 focuses on the detailed technical analysis and evaluation of the proposed method, while Section 5 provides a broader interpretation of the results, their implications, and their relevance to the field. This separation ensures that both aspects are thoroughly addressed without overlap, offering a more comprehensive and organized presentation of the work.

3. Check all abbreviations.

Answer: Concern has been addressed.

4. The paper should be submitted carefully to the template of the journal.

Answer: Concern has been addressed.

---

## [Editor Report · Decision Letter 1]

Where Octagonal Geometry Meets Chaos: A New S-Box for Advanced Cryptographic Systems

PONE-D-24-40373R1

Dear Dr. Iqbal,

We’re pleased to inform you that your manuscript has been judged scientifically suitable for publication and will be formally accepted for publication once it meets all outstanding technical requirements.

Kind regards,

Haider TH. Salim ALRikabi, Ph.D.

Academic Editor

PLOS ONE
---

## [Editor Report · Acceptance letter]

PONE-D-24-40373R1

PLOS ONE

Dear Dr. Iqbal,

I'm pleased to inform you that your manuscript has been deemed suitable for publication in PLOS ONE. Congratulations! Your manuscript is now being handed over to our production team.

Kind regards,

on behalf of

Dr. Haider TH. Salim ALRikabi

Academic Editor

PLOS ONE